# Prevalence of Childhood Overweight and Obesity in Liverpool between 2006 and 2012: Evidence of Widening Socioeconomic Inequalities

**DOI:** 10.3390/ijerph15122612

**Published:** 2018-11-22

**Authors:** Robert J. Noonan

**Affiliations:** Department of Sport and Physical Activity, Edge Hill University, Ormskirk L39 4QP, UK; Robert.Noonan@edgehill.ac.uk; Tel.: +44-1695-584-488

**Keywords:** children, deprivation, socioeconomic health inequalities, obesity, Liverpool

## Abstract

The primary aim of this study was to describe the prevalence of childhood overweight and obesity in Liverpool between 2006 and 2012. A secondary aim was to examine the extent to which socioeconomic inequalities relating to childhood overweight and obesity in Liverpool changed during this six-year period. A sample of 50,125 children was created using data from the National Child Measurement Program (NCMP) in Liverpool. The prevalence of overweight and obesity was calculated for Reception and Year 6 aged children in Liverpool for each time period by gender and compared against published averages for England. Logistic regression analyses examined the likelihood of children in Liverpool being classified as overweight and obese based on deprivation level for each time period. Analyses were conducted separately for Reception and Year 6 aged children and were adjusted for gender. The prevalence of overweight and obesity among Reception and Year 6 aged children in Liverpool increased between 2006 and 2012. During the same period, socioeconomic disparities in overweight and obesity prevalence between children living in the most deprived communities in Liverpool and those living in less deprived communities in Liverpool, widened. This study evidences rising rates of overweight and obesity among Liverpool children and widening socioeconomic health inequalities within Liverpool, England’s most deprived city between 2006 and 2012.

## 1. Introduction

Childhood obesity is a significant global public health concern [1]. In the UK and other developed countries, the prevalence of childhood obesity has increased markedly in recent decades [2]. Childhood obesity is a risk factor for type-2 diabetes and cardiovascular disease [3,4] as well as a strong predictor of mortality [5]. Socioeconomic inequalities in children’s health, including obesity, exist across England, with the burden of poor health falling disproportionately on children living in deprived communities [6]. Data from the UK Millennium Cohort Study evidenced a deprivation gradient to childhood obesity in England [7]. There are complex pathways linking socioeconomic disadvantage and childhood obesity. Evidence suggests that these socioeconomic health disparities are the result of differential access to economic and built environmental resources [8]. Deprived neighbourhoods are, for example, the most obesogenic (i.e., encourage the consumption of unhealthy food and/or limit opportunities for physical activity). They have the highest concentration of fast-food outlets [9,10] and provide the fewest opportunities for physical activity (e.g., being less walkable, limited access to self-contained gardens/yards, leisure facilities [6]). At the individual level, children from deprived neighbourhoods often live in low-income family households that may struggle to afford providing children a balanced diet [11] and active lifestyle [12]. These individual and area level factors act individually and collectively [13] and place children living in deprived neighbourhoods at greater risk of developing obesity compared with their more affluent peers [14,15].

Although the prevalence of childhood obesity in England and other developed countries has largely plateaued since the mid-2000s [16,17], children living in the most deprived areas of England have not benefited from this trend [18,19], and obesity rates remain high in these deprived communities [14]. Liverpool is the sixth largest English city, and it was ranked as the most deprived up until recently [20], with over 28% of Liverpool’s children living in poverty [21]. Research conducted in Liverpool almost a decade ago revealed that between 2003 and 2006, obesity rates among 9–10-year-old children had ‘stabilized’ [22]. At the time, Liverpool was on track to achieving the 2010 obesity partnership agreement of halting the year-on-year increase in childhood obesity [23].

Childhood obesity has no single one cause. However, since 2006, Liverpool has experienced a range of social and economic changes that may have, first, contributed to a rise in childhood obesity and, second, widened socioeconomic health inequalities. Following the onset of the economic recession in 2008, significant financial reductions were made to local government budgets (including Liverpool), which led to some public health promotion services (i.e., children’s play centres and leisure centres) being reduced, re-organised or, in some instances, curtailed [24]. Health promotion initiatives aimed at tackling childhood obesity in Liverpool have also since been terminated. These included, but are not limited to, initiatives such as the Liverpool City Council’s Active City [25] and Taste for Health strategies [26] and the SportsLinx project [27]. These social and economic changes have impacted at the individual family level too by decreasing household budgets (combination of rising food and fuel prices and stagnating wages and welfare benefits), which has led to fewer families (especially the most socially disadvantaged) being able to provide a balanced diet for children [28,29,30,31]. In this context, further research is warranted to understand whether the overweight and obesity trend witnessed in Liverpool between 2003 and 2006 continued post 2006.

The National Child Measurement Program (NCMP) is an annual program that measures the stature and body mass of children in Reception (aged 4–5 years) and Year 6 (aged 10–11 years) within state-maintained schools in England. The NCMP has been carried out since 2006. The present study uses NCMP data to (1) describe the prevalence of childhood overweight and obesity in Liverpool between 2006 and 2012 and (2) examine the extent to which socioeconomic inequalities relating to childhood overweight and obesity in Liverpool changed during this six-year period.

## 2. Materials and Methods

### 2.1. Participants

This study is a secondary analysis of NCMP data for Liverpool children. NCMP data were downloaded from the UK Data Service (https://www.ukdataservice.ac.uk), and a data set was constructed using data collected by Liverpool Primary Care Trust from the beginning of the NCMP in 2006–2007 to 2011–2012. Data were available for 51,359 children aged 4–11 years. Children attending independent and special schools were removed from the data set (*n* = 1234) for consistency with NCMP reports. This research involved secondary analysis of the NCMP and therefore did not require ethical approval.

### 2.2. Measures

#### 2.2.1. Childhood Overweight and Obesity

Stature and body mass were measured in schools by assessment teams recruited, trained and supervised by Local Authority public health departments [32]. Stature was measured to the nearest 0.1 cm using a portable stadiometer, and body mass was measured to the nearest 0.1 kg using calibrated scales. Body mass index (BMI) was calculated from stature and body mass as a proxy measure of body composition (kg/m^2^). The British 1990 growth reference (UK90) age-specific and gender-specific BMI cut-points were used to classify children as normal weight, overweight and obese [33]. Overweight and obesity are classified as above the 85th and 95th centile, respectively.

#### 2.2.2. Socioeconomic Inequality

Area-level deprivation was calculated from school postcodes using the 2010 English Indices of Multiple Deprivation (IMD; [34]) from the Department for Communities and Local Government and used as a measure of socioeconomic inequality. The IMD is a measure produced by the UK Government and comprises seven areas of deprivation: income, employment, health, education, housing, environment and crime. The IMD ranks each small area in England from the most to least deprived. Once areas have been ranked, they are then split into 10 equal-sized groups (deciles), ranging from the most (decile 1) to the least (decile 10) deprived areas nationally. Over 51% (*n* = 25,583) of the sample fell within the most deprived 10% of areas in England. Therefore, a categorical variable was created to represent children attending schools in the most deprived communities (decile 1) and less deprived communities (deciles 2–10).

### 2.3. Analysis

Changes in overweight and obesity among Liverpool Reception (aged 4–5 years) and Year 6 (aged 10–11 years) children were examined from 2006 to 2012 using three time periods: 2006–2008, 2008–2010 and 2010–2012. To analyze study aim 1, the mean percentages for overweight and obesity prevalence were calculated for Reception and Year 6 aged boys and girls in Liverpool for each time period. The mean percentage data for Reception- and Year 6-aged Liverpool children were then plotted in separate graphs against published mean percentage data for England [35,36,37,38,39,40] to improve interpretation. To address study aim 2, logistic regression analyses examined the likelihood of children in Liverpool being classified as overweight and obese based on deprivation level (i.e., most deprived vs. less deprived) for each time period. Analyses were conducted separately for Reception and Year 6 aged children and were adjusted for gender. The highest level of deprivation was the reference category. All analyses were conducted using Microsoft Excel 2016 (Microsoft, Redmond, WA, USA) and IBM SPSS v. 24 (SPSS Inc., Chicago, IL, USA), and statistical significance was set at *p* < 0.05.

## 3. Results

Data were available for 50,125 children (*n* = 25,638 boys; 51.15%) of which 51.68% were Reception aged children (*n* = 25,905) and 48.32% were Year 6 aged children (*n* = 24,220). In 2010–2012, 16,939 children participated (*n* = 9116 Reception age; 53.82%), compared with 16,603 children (*n* = 8539 Reception age; 51.43%) in 2008–2010 and 16,583 children (*n* = 8333 Year 6 age; 50.25%) in 2006–2008.

### 3.1. Study Aim 1

Figure 1a,b present the change in overweight and obesity prevalence between 2006 and 2012 among children in Liverpool and England. The prevalence of overweight (boys = 0.82%; girls = 0.83%; Figure 1a) and obesity (boys = 0.52%; girls = 0.06%) among Reception aged children in England declined between 2006 and 2012. A similar pattern of decline was observed between 2006 and 2010 for Reception aged children in Liverpool, but prevalence figures increased after 2010. Overall, the prevalence of overweight among Reception aged boys and girls in Liverpool increased by 2.25% and 2.56%, respectively. The prevalence of obesity among Reception aged boys (0.50%) and girls (0.90%) in Liverpool also increased during this period but less so.

The prevalence of overweight (boys = 1.37%; girls = 1.72%; Figure 1b) and obesity (boys = 1.15%; girls = 1.33%) among Year 6 aged children in England increased between 2006 and 2012. For Year 6 aged children in Liverpool, overweight and obesity prevalence increased by 2.60% and 2.61% among boys, respectively, and by 3.00% and 3.34% among girls, respectively.

### 3.2. Study Aim 2

In the period 2006–2012, Reception aged children living in the most deprived communities in Liverpool were consistently more likely to be overweight and obese compared with children living in less deprived communities in Liverpool (Figure 2). For Reception aged Liverpool children, disparities in the prevalence of overweight and obesity between children living in the most deprived communities and children living in less deprived communities increased linearly between 2006 and 2008 (OR = 1.07; *p* = 0.22; OR = 1.11; *p* = 0.13), between 2008 and 2010 (OR = 1.26; *p* < 0.001; OR = 1.20; *p* < 0.01), and between 2010 and 2012 (OR = 1.30; *p* < 0.001; OR = 1.29; *p* < 0.001), respectively.

In the period 2006–2012, Year 6 aged children living in the most deprived communities in Liverpool were consistently more likely to be overweight and obese compared with children living in less deprived communities in Liverpool (Figure 3). For Year 6 aged Liverpool children, disparities in the prevalence of overweight between children living in the most deprived communities and children living in less deprived communities, increased linearly between 2006 and 2008 (OR = 1.07; *p* = 0.14), between 2008 and 2010 (OR = 1.19; *p* < 0.001), and between 2010 and 2012 (OR = 1.32; *p* < 0.001). A similar trajectory was observed for childhood obesity in Liverpool, which increased between 2006 and 2008 (OR = 1.12; *p* < 0.05) and 2010 and 2012 (1.32; *p* < 0.001) but was greatest between 2008 and 2010 (OR = 1.34; *p* < 0.001).

## 4. Discussion

This study is the first to assess the prevalence of overweight and obesity among Liverpool children between 2006 and 2012 and examine whether socioeconomic inequalities relating to childhood overweight and obesity in Liverpool widened during this six-year period. Results revealed that deprivation had a consistent influence on childhood overweight and obesity in Reception and Year 6 aged children in Liverpool. For both age groups, disparities in the prevalence of overweight between children living in the most deprived communities in Liverpool and children living in less deprived communities in Liverpool increased linearly between 2006 and 2012. While a similar trend was evident for the prevalence of obesity among Reception aged Liverpool children, this was not evident for Year 6 aged Liverpool children. This finding evidences the importance of assessing childhood health and socioeconomic health inequalities at different ages. Moreover, the results demonstrate the need for continued investment in the NCMP in order to understand the future extent and impact of childhood obesity across demographic groups in England.

This study provides evidence that the prevalence of overweight and obesity among Liverpool Reception and Year 6 aged children increased between 2006 and 2012. Recent research based on nationally representative data collected in 2007–2008 evidenced a deprivation gradient to overweight and obesity among 7-year-old English children [7]. Children living in the most deprived communities of England were 1.61 times more likely to have overweight/obesity compared with children living in less deprived communities in England. The present study shows that Liverpool, which was the most deprived city in England at the time, has since then experienced rising childhood overweight and obesity rates. In the UK, it has been shown that obesity tracks strongly from childhood into adolescence [41]. Therefore, a substantial and increasing number of Liverpool children are at risk of developing obesity-related disorders in adolescence.

The main analyses revealed that socioeconomic inequalities relating to childhood overweight and obesity in Liverpool widened between 2006 and 2012. This is a significant finding as it demonstrates that in addition to growing disparities in socioeconomic health inequalities between English cities [18,19], inequalities have widened within the most deprived of English cities. It is important to note that Liverpool is a homogenous city with high levels of childhood poverty and obesity [21]. If compared with children living in more affluent areas of England, the socioeconomic inequalities relating to childhood overweight and obesity reported here may have been more marked [7]. In this study, socioeconomic inequalities relating to childhood overweight and obesity were evident at Reception age, which builds on current evidence relating to older children [14]. Based on these results, it would appear that more health initiatives are needed in the early years to reduce the onset of obesity at Reception age. Given that childhood obesity and socioeconomic health inequalities track strongly into adolescence [41], the findings of this study demonstrate the need for a new approach to reducing childhood obesity and tackling socioeconomic health inequalities in England to prevent socioeconomic health inequalities continuing into adolescence.

The results of this study show that between 2006 and 2012, strategies to combat childhood obesity and socioeconomic health inequalities in Liverpool did not work. This is concerning because between 2006 and 2010, the then Labour Government invested heavily in children-focused health programs to improve health and reduce socioeconomic health inequalities [42]. It is important to note that the data presented here may not accurately reflect the current magnitude of socioeconomic inequalities relating to childhood overweight and obesity in Liverpool. In the years following 2012, a range of additional austerity measures (including cuts to children and family support services, welfare benefits and tax credits) have been implement by the now Conservative Government, which have hit the poorest areas of the UK the hardest, most notably Liverpool [28,29,30]. Parallel to these welfare reforms and austerity programs, household budgets have further decreased in response to rising food and fuel prices and stagnating wages and welfare benefits [31]. In the UK, food is now 20% less affordable for the most socially disadvantaged families than in the mid-2000s [43]. Consequently, a majority of socially disadvantaged families are ‘food insecure’ (i.e., ability to access enough food for an active, healthy life) and rely on food charity (i.e., food banks) [44,45]. Liverpool has some of the highest levels of child poverty [46]. It is not unreasonable to suggest that the cumulative effect of these economic and social policies has resulted in fewer families in Liverpool being able to afford a healthy and balanced diet for their children.

With these current and future economic challenges in mind, it is important that the NCMP and programs alike are invested in and supported over the coming years so that the extent and impact of austerity on childhood obesity across demographic groups in England can be assessed. Further qualitative research is warranted to understand the effect of austerity and cuts to public health services on parents’ ability to support a healthy and active lifestyle for their family.

Evidence shows that to improve childhood health, reduce socioeconomic health inequalities and foster more equitable communities for health-promoting behaviours (i.e., physical activity and healthy eating), policy-level intervention is required [47,48]. However, for too long, the focus has been on behavioural interventions alone (individual responsibility) to tackle childhood obesity and socioeconomic health inequalities, with limited recognition of the policies and the social-environmental conditions that inform health-related decisions and behaviours. To demonstrate this point, Change4Life is a national social marketing campaign aimed at preventing childhood obesity by promoting healthy eating and physical activity [49]. At the same time, services aimed at supporting early childhood development and reducing socioeconomic health inequalities, such as Sure Start children’s centres, have lost roughly 40% [0.6 billion] of their funding since 2010 [50]. The largest cuts have occurred in the most disadvantaged local authorities (e.g., Liverpool), which are arguably the areas in greatest need. In the same period, the most disadvantaged communities have become more obesogenic [51] by way of indirectly encouraging the consumption of unhealthy food (e.g., high concentration of fast-food outlets and limited access to fresh produce). If we take Liverpool as an example, the city currently houses almost 700 fast-food outlets, a figure which has risen by roughly 10% since 2014 [51]. It is difficult to determine the impact these specific economic and built environmental changes have had on childhood obesity rates in Liverpool given the multifactorial nature of obesity. However, the message here rather relates to the need for greater synergy between government policies and public health priorities. In current contexts, efforts to reduce childhood obesity and socioeconomic health inequalities may be unrealistic.

This study has several strengths. The study is the first to examine changes in the prevalence of overweight and obesity in Liverpool children by deprivation level. The study sample was large, and socioeconomic health inequalities among children were examined for young and older ages. However, there are also some study limitations to consider. Although BMI is the most common measure of weight status in childhood [2], the measure reflects both fat and fat-free components of body mass [52] and has been known to underestimate excess body fat mass in children [53]. Moreover, overweight/obesity was defined using the UK90 criteria. While this reference is recommended [33,54,55] and has been used previously for population monitoring and clinical assessment of overweight/obesity in UK children [56,57], estimates of the prevalence of childhood overweight/obesity rely on the reference criteria used [58,59]. Therefore, the use of the UK90 criteria may have over/underestimated the prevalence of childhood overweight/obese in this study compared to other international criteria (e.g., Centre for Disease Control and Prevention, International Obesity Task Force and World Health Organization charts). Deprivation classifications were based on the geographical location of schools rather than home addresses, which may not have accurately reflected the actual deprivation level of all participating children. Furthermore, as the NCMP did not achieve 100% participation across all years (see References [35,36,37,38,39,40] for detail), the inconsistent participant response rates may have biased results by underestimating the prevalence of childhood overweight and obesity. For example, healthy weight children may have been more likely to participate in the study. English children of non-white ethnicity are at increased risk of obesity than their white peers [60], but data were unavailable to explore ethnic differences in overweight/obesity rates in the present study. However, given that the ethnic demography of Liverpool’s population is predominantly white [61], failure to include this data is unlikely to have impacted on the reported findings.

## 5. Conclusions

Children living in the most deprived communities in Liverpool are at greatest risk of overweight and obesity. This study evidences that between 2006 and 2012, the prevalence of childhood overweight and obesity in Liverpool increased. During this six-year period, socioeconomic inequalities relating to the prevalence of childhood overweight and obesity widened in Liverpool, England’s most deprived city at the time. The clear demonstration of a widening in socioeconomic health inequality among children in Liverpool is very significant in view of current ongoing austerity and public health funding cuts in England. Moreover, demonstration of the ineffectiveness of recent government policy, which specifically aimed to reduce socioeconomic health inequalities among children in England, evidences the need to maintain strong focus on strengthening the current Childhood Obesity Plan [62]. A new approach to reducing childhood obesity and tackling socioeconomic health inequalities in England is needed.

## Figures and Tables

**Figure 1 ijerph-15-02612-f001:**
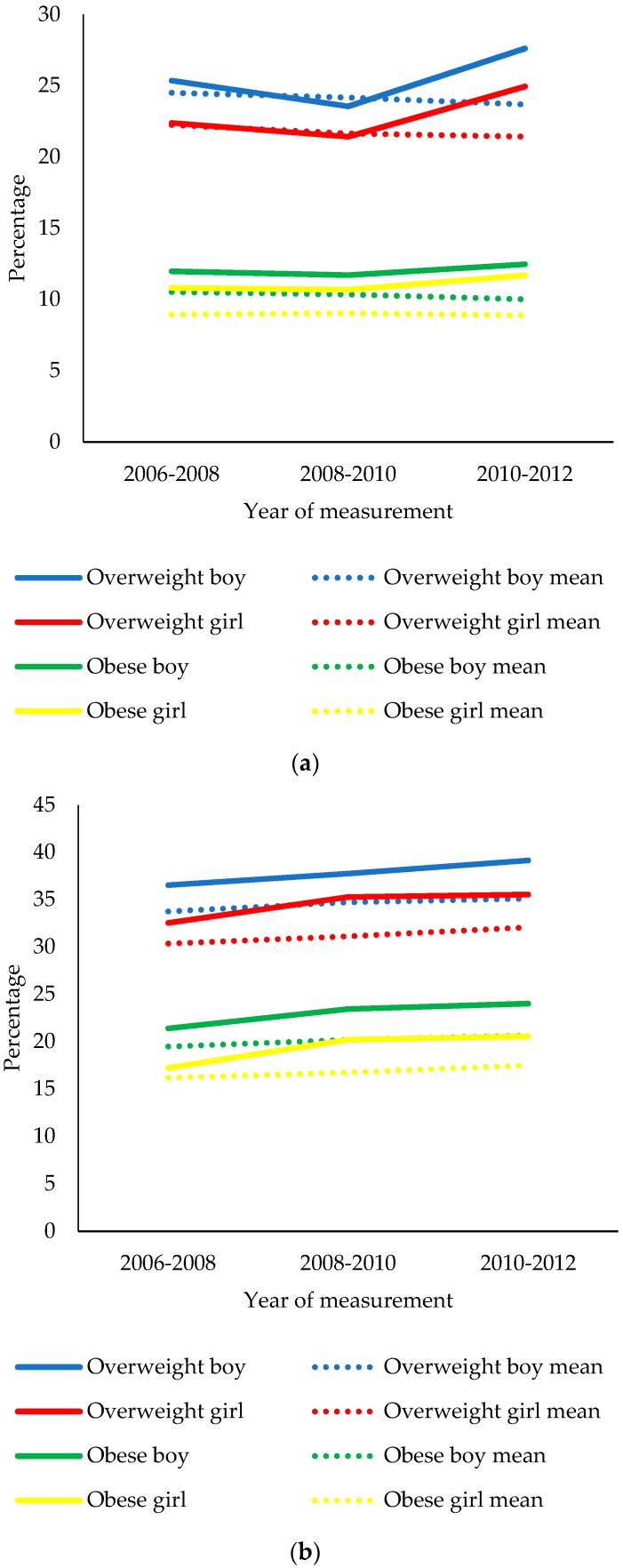
(**a**) Change in the prevalence of overweight and obesity among Reception aged children in Liverpool and England between 2006 and 2012. Mean percentage values for overweight and obese boys and girls = English average. (**b**) Change in the prevalence of overweight and obesity among Year 6 aged children in Liverpool and England between 2006 and 2012. Mean percentage values for overweight and obese boys and girls = English average.

**Figure 2 ijerph-15-02612-f002:**
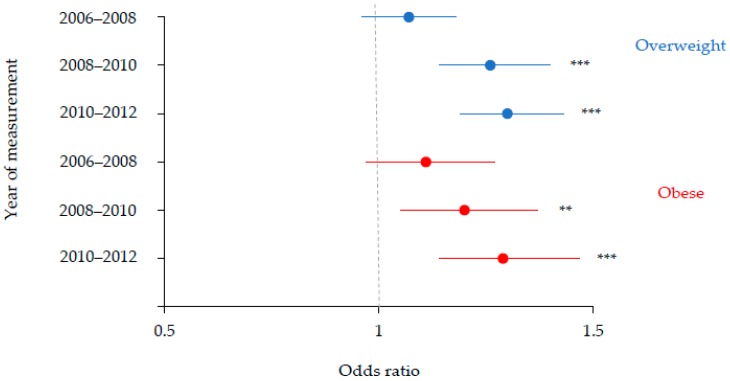
Odds ratios with 95% confidence intervals illustrating associations between deprivation and weight status among Reception aged Liverpool children between 2006 and 2012. The highest decile of deprivation was the reference category. Adjusted for gender. ** *p* < 0.01; *** *p* < 0.001.

**Figure 3 ijerph-15-02612-f003:**
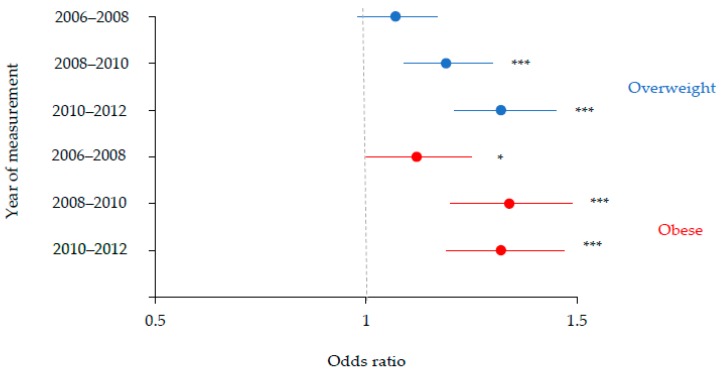
Odds ratios with 95% confidence intervals illustrating associations between deprivation and weight status among Year 6 aged Liverpool children between 2006 and 2012. The highest decile of deprivation was the reference category. Adjusted for gender. * *p* < 0.05; *** *p* < 0.001.

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
