# Peer review of "Prevalence of Childhood Overweight and Obesity in Liverpool between 2006 and 2012: Evidence of Widening Socioeconomic Inequalities"

_ijerph, 2018, doi:10.3390/ijerph15122612_

Round 1
Reviewer 1 Report
This manuscript compares the prevalence of overweight and obesity in children from Liverpool in 2006-2012. This is the sixth largest city in the UK, where the prevalence of excess weight has increased greatly in the last decades. Also, this paper tries to find an association between the risk of being overweight/obese and deprivation, since Liverpool has about 30% of the children population living in poverty. In the context of non-communicable chronic diseases, the relationship of health status and inequality has been largely explore in developing nations; less is known about this topic in developed countries, particularly in Western European countries, where income distribution is better compared to the US and other mid-to-high income nations. Thus, the topic is of great interest for public health purposes. The author analyses data from the National Child Measurement Programme (NCMP) in Liverpool, reaching a sample size of 50,000 preschool and schoolchildren. The paper is well written and I pretty much enjoyed doing the review. However, I have some suggestions, which are intended to improve the study.
Introduction
Line 28 >> Please replace type 11 diabetes with type-2 diabetes
Line 53-55 >> I wonder whether you have any hypothesis on how deprivation levels or inequality affects the risk of being obese. What does the literature say about this issue in countries with relatively good equality indexes?
Methods
Line 67-73 >> Weight status is assessed with the 1990 growth reference in the UK. Thus, overweight and obesity are classified as above the 85th and 95th centile, respectively. I suggest replicating the analysis with the 2007 WHO reference and see whether the relationships obtained with the UK90 remains significant with the WHO child growth standard. I’d like to see both analyses.
Line 83-84 >> I’d greatly appreciate if you could clarify the age rage for Reception and Year 6 age children; I’m not from the UK and thus I had to take look in the web to find out that Reception children are preschooler and Year 6 age children are 6th graders. It’s not very intuitive.
Data analysis
Is it possible that age or school level and study wave are entered as control variables in the models testing the relationship of inequality and risk of excess weight? The author may want to present OR before and after adjustments to see whether the relationship is independent of these factors.
I’d like to know whether a formal statistical tests was used to found differences in the trend of obesity /overweight in preschoolers and sixth graders over the 2006-2012 period. Also, to find differences in the observed trend in Liverpool compared to the national trend. I think something more powerful than the graphic test is needed to reach a conclusion. Graphs should be used as a complement of formal statistical tests and I strongly encourage the author to look for something more analytical.
Author Response
I appreciate the time and efforts by the editor and referees in reviewing the manuscript. The comments provided have been most helpful. I have addressed all concerns and comments indicated in the review reports, and as a result believe that the revised version is much improved in line with the journal publication requirements.
1. Line 28 >> Please replace type 11 diabetes with type-2 diabetes
Type-2 diabetes has been included on line 31.
2. Line 53-55 >> I wonder whether you have any hypothesis on how deprivation levels or inequality affects the risk of being obese. What does the literature say about this issue in countries with relatively good equality indexes?
As discussed in the opening section of the introduction on line 33-37, UK evidence shows that children living in areas of high deprivation are at greater risk of obesity compared to their more affluent peers (Noonan & Fairclough, 2018; Noonan et al. 2016). From a neighbourhood deprivation perspective, evidence suggests that deprived neighbourhoods are more likely to be obesogenic (i.e., encourage the consumption of unhealthy food and/or limit opportunities for physical activity). Several studies have showed that deprived neighborhoods have a greater concentration of fast-food outlets (Cetateanu & Jones, 2014; Maguire et al. 2015), are less walkable (Noonan et al. 2016) and have limited access to self-contained gardens/yards compared with affluent neighborhoods (Noonan et al. 2016). Such characteristics place children living in deprived neighborhoods at greater risk of developing obesity compared with their more affluent peers (Conrad & Capewell, 2012; Goisis et al. 2015).
3. Line 67-73 >> Weight status is assessed with the 1990 growth reference in the UK. Thus, overweight and obesity are classified as above the 85th and 95th centile, respectively. I suggest replicating the analysis with the 2007 WHO reference and see whether the relationships obtained with the UK90 remains significant with the WHO child growth standard. I’d like to see both analyses.
I acknowledge the point raised by the reviewer. In England the British 1990 growth reference (UK90) is recommended for population monitoring and clinical assessment in children aged four years and over (Cole, Freeman & Preece, 1995; 1998; Freeman et al. 1995). Extensive studies have used the reference charts to classify child weight status in the UK (Ellis et al. 2015; Stamatakis et al. 2005). I feel that studying whether relationships between deprivation and weight status were consistent across reference charts would detract from the main focus of the paper. The analyses proposed by the reviewer are similar to those conducted previously (Gonzalez-Casanova et al. 2013; Lang et al. 2011). I see the additional analyses contributing little to the current knowledge base.
4. Line 83-84 >> I’d greatly appreciate if you could clarify the age rage for Reception and Year 6 age children; I’m not from the UK and thus I had to take look in the web to find out that Reception children are preschooler and Year 6 age children are 6th graders. It’s not very intuitive.
The age range of Reception (4-5 years) and Year 6 children (10-11 years) is presented on line 52. The same information has now been included in the methods section on line 88-89.
5. Is it possible that age or school level and study wave are entered as control variables in the models testing the relationship of inequality and risk of excess weight? The author may want to present OR before and after adjustments to see whether the relationship is independent of these factors.
Analyses were conducted separately for age groups because of the large discrepancies in overweight/obesity prevalence between Reception and Year 6 children. The decision was also informed by prior research in this area which has shown a strong positive association between age and obesity prevalence (Bann et al. 2018; Hughes et al. 2011; Jabakhanji et al. 2018; Moss & Yeaton, 2012; van Vliet et al. 2015). With regards to school level data, unfortunately, the data set did not present a variable reflective of individual schools which limits efforts to study the study outcomes at the school level. Instead, only the deprivation decile was presented. The decision to adjust the main analyses [logistic regressions] for gender was informed by prior research in this area in which gender influenced child overweight/obesity rates (Lo et al. 2014; Noonan, 2018; Noonan & Fairclough, 2018; van Vliet et al. 2015; Wisniewski & Chernausek, 2009).
6. I’d like to know whether a formal statistical tests was used to found differences in the trend of obesity /overweight in preschoolers and sixth graders over the 2006-2012 period. Also, to find differences in the observed trend in Liverpool compared to the national trend. I think something more powerful than the graphic test is needed to reach a conclusion. Graphs should be used as a complement of formal statistical tests and I strongly encourage the author to look for something more analytical.
The reviewer raises an important point. On reflection, this statement is misleading to the reader. Statistical tests were not conducted between Liverpool and the national average because of the large discrepancies in sample sizes. Within aim 1, the terminology used has been revised to match the analyses conducted and results presented. The term “describe” is used instead of “compare” on line 9 and line 54. The main aim of the present study was to demonstrate the widening of difference in overweight/obesity prevalence across the years studied in Liverpool.
References
Bann, D.; Johnson, W.; Li, L.; Kuh, D.; Hardy, R. Socioeconomic inequalities in childhood and adolescent body-mass index, weight, and height from 1953 to 2015: An analysis of four longitudinal, observational, British birth cohort studies. Lancet Public Health 2018, 3, e194–e203
Cetateanu A.; Jones A. Understanding the relationship between food environments, deprivation and childhood overweight and obesity: evidence from a cross sectional England-wide study. Health Place 2014, 27, 68–76.
Cole T.J.; Freeman J.V.; Preece M.A. Body mass index reference curves for the UK, 1990. Archives of Disease in Childhood 1995, 73, 25‐29.
Cole T.J.; Freeman J.V.; Preece M.A. British 1990 growth reference centiles for weight, height, body mass index and head circumference fitted by maximum penalized likelihood. Statistics in Medicine 1998, 17, 407‐29.
Conrad D.; Capewell S. Associations between deprivation and rates of childhood overweight and obesity in England, 2007–2010: an ecological study. BMJ Open 2012, 2, e000463.
Ells L.J.; Hancock C.; Copley V.R.; et al Prevalence of severe childhood obesity in England: 2006–2013 Archives of Disease in Childhood 2015, 100, 631-636.
Freeman J.V.; Cole T.J.; Chinn S.; Jones P.R.M.; White E.M.; Preece M.A. Cross sectional stature and weight reference curves for the UK, 1990. Archives of Disease in Childhood 1995;73, 17‐24.
Goisis G.; Sacker A.; Kelly Y. Why are poorer children at higher risk of obesity and overweight? A UK cohort study. Eur J Pub Health 2015, 26, 7–13.
Gonzalez-Casanova I.; Sarmiento O.L.; Gazmararian J.A.; Cunningham S.A.; Martorell R.; Pratt M.; Stein A.D. Comparing three body mass index classification systems to assess overweight and obesity in children and adolescents. Rev Panam Salud Publica 2013, 33, 349-355.
Hughes A.R.; Sherriff, A.; Lawlor D.A.; Ness A.R.; Reilly J.J. Incidence of obesity during childhood and adolescence in a large contemporary cohort. Prev Med 2011, 52, 300-304.
Jabakhanji S.B.; Boland F.; Ward M.; Biesma R. Body Mass Index Changes in Early Childhood. J Pediatr 2018, 202, 106-114.
Lang I.A.; Kipping R.R.; Jago R.; Lawlor D.A. Variation in childhood and adolescent obesity prevalence defined by international and country-specific criteria in England and the United States. Eur J Clin Nutr 2011, 65,143-150.
Lo J.C.; Maring B.; Chandra M.; et al. Prevalence of obesity and extreme obesity in children aged 3-5 years. Pediatr Obes 2014, 9, 167-175.
Maguire E.R.; Burgoine T.; Monsivais P. Area deprivation and the food environment over time: A repeated cross-sectional study on takeaway outlet density and supermarket presence in Norfolk, UK, 1990–2008. Health & Place 2015, 33, 142-147.
Moss B.G.; Yeaton W.H. U.S. children's preschool weight status trajectories: patterns from 9-month, 2-year, and 4-year Early Childhood Longitudinal Study-Birth cohort data. Am J Health Promot 2012, 26,172-175.
Noonan R.J. The effect of childhood deprivation on weight status and mental health in childhood and adolescence: longitudinal findings from the Millennium Cohort Study. Journal of Public Health 2018,
Noonan, R.J.; Boddy, L.M.; Knowles, Z.R.; Fairclough, S.J. Cross-sectional associations between high-deprivation home and neighbourhood environments, and health-related variables among Liverpool children. BMJ Open. 2016, 6, e008693.
Noonan, R.J.; Fairclough, S.J. Is there a deprivation and maternal education gradient to child obesity and moderate-to-vigorous physical activity? Findings from the Millennium Cohort Study. Pediatr Obes. 2018, 13, 458-64.
Stamatakis E.; Primatesta P.; Chinn S.; Rona R.; Falascheti E. Overweight and obesity trends from 1974 to 2003 in English children: what is the role of socioeconomic factors? Archives of Disease in Childhood 2005, 90, 999–1004.
van Vliet J.S.; Gustafsson P.A.; Duchen K.; Nelson N. Social inequality and age-specific gender differences in overweight and perception of overweight among Swedish children and adolescents: a cross-sectional study. BMC Public Health 2015, 15, 628.
Wisniewski A.B.; Chernausek S.D. Gender in childhood obesity: Family environment, hormones, and genes. Gender Medicine 2009, 6, 76-85.
Reviewer 2 Report
Thank you for your efforts in submitting this interesting work. I have reviewed the paper and provide some comments to help improve the clarity and impact of the article.
General Comments
Would not removing students from independent and special schools provide a unique socio-economic and/or educational analysis rather than just comparing to average UK data? It could be that children attending these schools have different rates, irrespective of zip code zone classification of deprived living areas.
Why not also use level of deprivation as a covariate. I would assume that even within this classification there may be degrees of economic deprivation. It appears you ranked these but more description of rationale and how this was done is needed in the methods.
As mentioned in your limitations, you base the analysis on zip codes which is a limitation. Why not also include other factors that may influence inequities such as race/ethnicity, citizenship, family income, etc. I realize you are using an existing data base but the more data available that can add to interpretation would be helpful, if available. If it's not available, listing some of these additional factors in the limitation section might help others construct better data bases.
Add SD's or 95% CI's (preferred) to figures (reformat if necessary). Also, since you are interested in change over time, why not present as mean changes from baseline with 95% CI’s so you can determine if the change significantly differed over time (i.e., mean change and 95% CI completely above or below baseline). This could be a similar figure like the OR figure.
It’s also unclear from the figures and text whether you are just making observations about mean changes in percentages or whether these were statistically significant over time or among groups.
Figures should be fully interpretable. It may also be helpful to show these data in a table format with SD’s or 95% CI’s and all statistical findings.
Your odd’s ratio data is the most compelling. Are the bars SD, SEM, 95% CI’s, ranges? This needs to be defined on the figure and included in data presented in the text.
All methods (and calculations) need to described in the methods with supporting references. For example, I don’t see where you describe how you calculated the magnitude of inequality.
I’m also lost as to how the analysis was compared to average UK values. Were odds ratios normalized to that data? If so, this needs to be clearer in methods.
Your discussion is at times highly speculative with no direct data to support. For example, the lines 172 – 187 suggest changes are due to government funding issues, yet there is no data presented at all that established that programs cut directly impacted these children such that it would explain trends observed. It is just as likely other factors may have affected this (e.g., technology, gaming, neighborhood safety issues, lack of family support to promote healthy diet and physical activity, cultural issues, availability of healthy food, built environment to promote physical activity, etc.). In this case, you present no demographic data other than sex. Many urban neighborhoods experience demographic shifts as communities migrate to other areas in or outside of the city and new communities replace them. Were there shifts in ethnicity in these communities? More data is needed to make these types of comments.
The next paragraph is more evidence based and better referenced. However, you didn’t mention increasing physical activity in school settings or after school. Again, the authors are making a lot of assumptions with very little data presented (i.e., age, gender, BMI, and zip coded / economic zones). For example, one of the greatest impacts on lack of physical activity from 2006 – 2012 was technology (e.g., iPhones, social media, gaming, etc.). That would have also seeming made a major impact during this time period.
Thank you for your efforts to examine these important health equity issues.
Author Response
I appreciate the time and efforts by the editor and referees in reviewing the manuscript. The comments provided have been most helpful. I have addressed all concerns and comments indicated in the review reports, and as a result believe that the revised version is much improved in line with the journal publication requirements.
1. Would not removing students from independent and special schools provide a unique socio-economic and/or educational analysis rather than just comparing to average UK data? It could be that children attending these schools have different rates, irrespective of zip code zone classification of deprived living areas.
The decision to remove these schools from the Liverpool sample was informed by the criteria used to calculate prevalence figures in the national sample. This information is presented in the methods section on line 64-65.
2. Why not also use level of deprivation as a covariate? I would assume that even within this classification there may be degrees of economic deprivation. It appears you ranked these but more description of rationale and how this was done is needed in the methods.
The decision to not include deprivation as a covariate in the analyses was informed by recent published work in this area (Noonan, 2018a; 2018b; Noonan & Fairclough, 2018; Noonan et al. 2016). The focus of the paper was on deprivation, and deprivation was the independent variable in all main analyses.
The IMD ranks each small area in England from the most to least deprived. Once areas have been ranked, they are then split into ten equal-sized groups (deciles), ranging from the most (decile 1) to least deprived areas nationally (decile 10). Over 51% (n = 25583) of the sample fell within the most deprived 10% of areas in England. Therefore, a categorical variable was created to represent children attending schools in the most deprived communities (decile 10) and less deprived communities (deciles 1 - 9). This additional detail has now been provided on line 81-84.
3. As mentioned in your limitations, you base the analysis on zip codes which is a limitation. Why not also include other factors that may influence inequities such as race/ethnicity, citizenship, family income, etc. I realize you are using an existing data base but the more data available that can add to interpretation would be helpful, if available. If it's not available, listing some of these additional factors in the limitation section might help others construct better data bases.
I thank the reviewer for raising this point. As noted, the data was derived from an existing data base. It was my intention to use individual level zip codes, but this level of detail was not collected in the National Child Measurement Programme. It would have also been useful to report data on ethnicity. However, this data was not present in the data set. As the ethnic demography of Liverpool’s population is predominantly white (Liverpool City Council, 2011), it is unlikely that ethnicity impacted greatly on the study findings. Further discussion related to these issues has been presented on line 239-242.
4. Add SD's or 95% CI's (preferred) to figures (reformat if necessary). Also, since you are interested in change over time, why not present as mean changes from baseline with 95% CI’s so you can determine if the change significantly differed over time (i.e., mean change and 95% CI completely above or below baseline). This could be a similar figure like the OR figure.
The error bars in figure 2 and 3 represent 95% confidence intervals. This information has now been included on line 130 and line 141. I appreciate the point raised by the reviewer. However, the derived national level data for England does not present SD’s or 95% CI and therefore we are not able to present this information.
5. It’s also unclear from the figures and text whether you are just making observations about mean changes in percentages or whether these were statistically significant over time or among groups.
The comparisons between Liverpool and the national average were based on mean changes in percentages. No statistical tests were conducted to determine whether these observed differences were statistically significant. I have included more explicit information in the analysis section of the manuscript to make this clearer to the reader. This information is presented on line 90-94.
6. Figures should be fully interpretable. It may also be helpful to show these data in a table format with SD’s or 95% CI’s and all statistical findings.
I appreciate the point raised by the reviewer regarding statistical significance. Statistical significance information has now been included in figure 2 and 3. The footnote beneath figure 2 and 3 has also been revised detailing each statistical significance level and corresponding asterix(s) on line 138-140 and line 151-153.
7. Your odd’s ratio data is the most compelling. Are the bars SD, SEM, 95% CI’s, ranges? This needs to be defined on the figure and included in data presented in the text.
The error bars in figure 2 and 3 represent 95% confidence intervals. This information has now been included on line 138 and line 151.
8. All methods (and calculations) need to described in the methods with supporting references. For example, I don’t see where you describe how you calculated the magnitude of inequality.
I thank the reviewer for raising this important point. I appreciate that this information is not explicitly clear to the reader. There were no statistical tests conducted to determine the magnitude of inequality. Rather, this observation was made based on the larger odds ratios observed during more recent years. These statements have been revised to improve clarity for the reader. Revisions have been made to line 18-20, line 132-133, line 144-145, line 147 and line 158-160.
9. I’m also lost as to how the analysis was compared to average UK values. Were odds ratios normalized to that data? If so, this needs to be clearer in methods.
Analyses between Liverpool and the UK were only conducted for study aim 1. In this case, it was percentage data that were compared descriptively. For study aim 2, only Liverpool data was used. I have included “Liverpool” more frequently in discussion to make explicitly clear to the reader that these [logistic regression/odds ratio] comparisons relate to Liverpool children alone rather than Liverpool vs national average comparisons. A more clearer description of the analyses has been provided on line 88-94.
10. Your discussion is at times highly speculative with no direct data to support. For example, the lines 172 – 187 suggest changes are due to government funding issues, yet there is no data presented at all that established that programs cut directly impacted these children such that it would explain trends observed. It is just as likely other factors may have affected this (e.g., technology, gaming, neighborhood safety issues, lack of family support to promote healthy diet and physical activity, cultural issues, availability of healthy food, built environment to promote physical activity, etc.). In this case, you present no demographic data other than sex. Many urban neighborhoods experience demographic shifts as communities migrate to other areas in or outside of the city and new communities replace them. Were there shifts in ethnicity in these communities? More data is needed to make these types of comments.
I recognize that the information presented in this section of the manuscript is not fully supported with evidence. The narrative has been changed accordingly to align with the evidence available. For example, there is strong evidence to suggest that the recently implemented economic and social policies have led to fewer people being able to afford a healthy and balanced diet (Lambie-Mumford, 2018; Lambie-Mumford & Green, 2017; Loopstra et al. 2015). Additional evidence has been presented on line 189-204.
According to Census data published in 2011, Liverpool’s population increased by 26,942 (6.1%) between 2001 and 2011 (Liverpool City Council, 2011). This was a smaller increase than the national rate of increase (7.8%). With regards to Liverpool’s ethnic population, Liverpool is less ethnically diverse than the population of England and Wales as a whole. In Liverpool 13.8% of the population are BME compared to 18.6% of the population nationally (Liverpool City Council, 2011b). Liverpool’s BME population increased significantly between 2001 and 2011, increasing by 33,700 people (Liverpool City Council, 2011b). However, between 2006-2012 Liverpool remained the most deprived local authority in England. This was the main focus of the manuscript (Department for Communities and Local Government, 2010).
11. The next paragraph is more evidence based and better referenced. However, you didn’t mention increasing physical activity in school settings or after school. Again, the authors are making a lot of assumptions with very little data presented (i.e., age, gender, BMI, and zip coded / economic zones). For example, one of the greatest impacts on lack of physical activity from 2006–2012 was technology (e.g., iPhones, social media, gaming, etc.). That would have also seeming made a major impact during this time period.
I have acknowledged the point raised by the reviewer and have revised this section of the manuscript to ensure the information presented is empirically informed. The section on leisure centre closures and physical activity has been removed [line 223] as there is insufficient evidence to demonstrate that cuts to leisure services has impacted on physical activity participation rates and contributed to the rise in child obesity. It has also been made clearer to the reader that obesity is multifactorial and as such it is difficult to determine the specific impact the recent economic and built environmental changes have had on child obesity rates in Liverpool. The narrative of this section has been revised to reflect this [line 211-228].
References
Department for Communities and Local Government. The English indices of deprivation 2010. Wetherby: Communities and Local Government Publications, 2010.
Lambie-Mumford H. The growth of food banks in Britain and what they mean for social policy. Critical Social Policy, 1–20.
Lambie-Mumford H.; Green MA. Austerity, welfare reform and the rising use of food banks by children in England and Wales. Area 2017, 49, 273–279.
Liverpool City Council. Liverpool’s Population: 2011 Census. https://liverpool.gov.uk/media/9905/population.pdf
Liverpool City Council Ethnicity in Liverpool: 2011 Census. https://liverpool.gov.uk/media/9899/ethnicity-and-migration.pdf
Loopstra R.; Reeves A.; Taylor-Robinson D.; Barr B.; McKee M. Austerity, sanctions, and the rise of food banks in the UK. BMJ 2015, 350, h1775.
Noonan R.J. The effect of childhood deprivation on weight status and mental health in childhood and adolescence: longitudinal findings from the Millennium Cohort Study. Journal of Public Health 2018,
Noonan R.J. Poverty, weight status and dietary intake among UK adolescents. International Journal of Environmental Research and Public Health 2018, 15, 1224.
Noonan RJ.; Fairclough SJ. Is there a deprivation and maternal education gradient to child obesity and moderate-to-vigorous physical activity? Findings from the Millennium Cohort Study. Pediatric Obesity 2018, 13, 458-464
Noonan R.J.; Boddy L.M.; Knowles Z.R.; Fairclough S.J. Cross-sectional associations between high-deprivation home and neighbourhood environments, and health-related variables among Liverpool children. BMJ Open 2016, 6, e008693.
Reviewer 3 Report
Contextualizing the impact of communal deprivation has on childhood weight statuses is a welcomed perspective to this research topic. This study is well organized and the methodology is well written. In my opinion the study outcomes offers enhanced evidence-based understanding of the prevalence of child overweight and obesity. As acknowledged by the authors, this Liverpool study is unique because it not only assesses the prevalence of overweight and obesity among Liverpool children, but it examined whether inequalities in child overweight and obesity in Liverpool widened during a 6-year period (e.g. 2006-2012).
Communal deprivation is a communal determinant of health and perhaps it puts children and adolescences at an even greater risk of overweight and obesity than previously acknowledged in the research.
The manuscript’s topic is very compelling and warrants publication.
Specific Recommendations
1. Even though the below recommendations a duly affirmed in the references, I suggest that you edit the Measures Section 2.2 on Page 2:
a. Add a footnote to be more transparently about the BMI measurement training the personnel completed for the study.
b. Add a sentence to address the validation of the English Indices of Multiple Deprivation (IMD) measurement tool
Author Response
I appreciate the time and efforts by the editor and referees in reviewing the manuscript. The comments provided have been most helpful. I have addressed all concerns and comments indicated in the review reports, and as a result believe that the revised version is much improved in line with the journal publication requirements.
Even though the below recommendations a duly affirmed in the references, I suggest that you edit the Measures Section 2.2 on Page 2:
1. Add a footnote to be more transparently about the BMI measurement training the personnel completed for the study.
Anthropometric measurements were conducted by trained healthcare professionals in schools. This information has been included on line 70. All personnel were shown how to conduct accurate height and weight measurements.
2. Add a sentence to address the validation of the English Indices of Multiple Deprivation (IMD) measurement tool.
The English Indices of Multiple Deprivation (IMD) is a UK Government produced measure of relative deprivation for every Local Authority District in England. The IMD aims to provide a nationally consistent measure of how deprived an area is by identifying the degree to which people are disadvantaged by factors such as low income, unemployment, lack of education, poor health, and crime. The measure is commonly used and is an accepted measure of area-level deprivation in child health research in the UK (Conrad & Capewell, 2012; Lloyd et al. 2017; Noonan, 2018a; 2018b; Noonan & Fairclough, 2018; Noonan et al. 2016; 2017; Pearce Webb-Phillips, Bray, 2016; Sebire et al. 2011; Taylor et al. 2017).
References
Conrad D.; Capewell S. Associations between deprivation and rates of childhood overweight and obesity in England, 2007–2010: an ecological study. BMJ Open 2012, 2, e000463.
Lloyd J.; Creanor S.; Logan S.; et al. Effectiveness of the Healthy Lifestyles Programme (HeLP) to prevent obesity in UK primary-school children: a cluster randomised controlled trial. Lancet Child Adolesc Health 2017, 2, 35–45.
Noonan R.J. The effect of childhood deprivation on weight status and mental health in childhood and adolescence: longitudinal findings from the Millennium Cohort Study. Journal of Public Health 2018a,
Noonan R.J. Poverty, weight status and dietary intake among UK adolescents. International Journal of Environmental Research and Public Health 2018b, 15, 1224.
Noonan RJ.; Fairclough SJ. Is there a deprivation and maternal education gradient to child obesity and moderate-to-vigorous physical activity? Findings from the Millennium Cohort Study. Pediatric Obesity 2018, 13, 458-464.
Noonan R.J.; Boddy L.M.; Knowles Z.R.; Fairclough S.J. Fitness, fatness and active school commuting among Liverpool Schoolchildren. International Journal of Environmental Research and Public Health 2017, 14, 995.
Noonan R.J.; Boddy L.M.; Knowles Z.R.; Fairclough S.J. Cross-sectional associations between high-deprivation home and neighbourhood environments, and health-related variables among Liverpool children. BMJ Open 2016, 6, e008693.
Pearce M.; Webb-Phillips S.; Bray I. Changes in objectively measured BMI in children aged 4–11 years: data from the National Child Measurement Programme. Journal of Public Health 2016, 38, 3, 459–466.
Sebire S.J.; Jago R.; Fox K.R.; Page A.S.; Brockman R.; Thompson J. L. Associations between children's social functioning and physical activity participation are not mediated by social acceptance: a cross-sectional study. International Journal of Behavioral Nutrition and Physical Activity 2011, 8, 106.
Taylor S.L.; Curry W.B.; Knowles Z.R.; Noonan R.J.; McGrane B.; Fairclough S.J. Predictors of segmented school day physical activity and sedentary time in children from a northwest England low-income community. International Journal of Environmental Research and Public Health 2017, 14, 534.
Round 2
Reviewer 1 Report
I appreciate the submission of an improved version of the article entitled ‘Overweight and obesity prevalence among Liverpool children, 2006-2012: Evidence for widening socioeconomic inequalities’. Like I said in my first review report, the topic addressed by the author (the relationship between inequality levels and overnutrition in children and adolescents) is one of outmost importance and, thus, his findings have the potential to inform preventive strategies tackling with obesity among younger age populations. In spite of the effort to provide a revised version of the manuscript, I still feel that paper is not ready for publication.
Here go my comments:
1. I still miss a well-formulated hypothesis; it is not enough to suggest a possible relationship between variables. In addition, in his response, the author refers to a series of research findings that he does not comment on in the Introduction or the Discussion. This must to be improved.
2. It seems reasonable to me that the author prefers the use of a national reference to evaluate the nutritional status of the sample. However, if he does not wanna use the more universal 2006 and 2007 WHO references he should at least do the following: (1) indicate this as a potential weakness of his analysis, and (2) comment his findings in light of the results obtained by Lang and González-Casanova.
3. The variable 'age group' may behave as a variable that denotes the school level of participants (preschoolers vs. 1st to 4th graders), so I accept the author’s decision as a valid one. However, even within the age groups there is variability in the age of participants. Then, I want to know if within groups age is a confounder in the relationship of interest, since younger children tend to be more protected by the health system and their families.
4. Even a descriptive analysis should be based on statistical grounds. There must be something more solid than pure observation of trajectories in a graph to draw conclusions. I want to see some statistical work here.
Author Response
I appreciate the time and efforts by the editor and reviewer in reviewing the manuscript. The comments have helped strengthen the overall narrative of the manuscript. I am thankful this feedback. I have addressed all concerns and comments indicated in the review report.
1. I still miss a well-formulated hypothesis; it is not enough to suggest a possible relationship between variables. In addition, in his response, the author refers to a series of research findings that he does not comment on in the Introduction or the Discussion. This must to be improved.
I have provided additional detail in the opening section of the manuscript to make it clearer to the reader why children living in deprived neighborhoods may be at greatest risk of obesity [line 34-45]. Moreover, to set the scene for the information presented in the discussion section of the manuscript, I have provided additional information in the latter part of the introduction section of the manuscript relating to the social and economic changes experienced in Liverpool during the period studied (i.e., 2006 - 2012) [line 54-67].
2. It seems reasonable to me that the author prefers the use of a national reference to evaluate the nutritional status of the sample. However, if he does not want to use the more universal 2006 and 2007 WHO references he should at least do the following: (1) indicate this as a potential weakness of his analysis, and (2) comment his findings in light of the results obtained by Lang and González-Casanova.
I have acknowledged the point raised by the reviewer and have provided additional discussion in the limitations section of the manuscript on line 252-258.
3. The variable 'age group' may behave as a variable that denotes the school level of participants (preschoolers vs. 1st to 4th graders), so I accept the author’s decision as a valid one. However, even within the age groups there is variability in the age of participants. Then, I want to know if within groups age is a confounder in the relationship of interest, since younger children tend to be more protected by the health system and their families.
I appreciate the initial information presented in the methods section of the manuscript may have been misleading to the reader. Therefore, the sentence on line 17-18 and line 111-112 has been revised to make known to the reader that analyses were conducted separately for Reception and Year 6 children. Unfortunately, the data set did not provide information on the participant’s decimal age. Instead, participants were sorted by year group (Reception or Year 6). I have therefore been unable to explore whether variability existed within age groups.
4. Even a descriptive analysis should be based on statistical grounds. There must be something more solid than pure observation of trajectories in a graph to draw conclusions. I want to see some statistical work here.
In order to remove this contention, comparisons have not been made between Liverpool and English averages. Statements denoting overweight/obesity differences between Liverpool and England have been deleted to better reflect the analyses conducted. Deletions have been made to line 10, line 19, line 71, line 22-24, line 134-136, line 187-188, and line 272.
Reviewer 2 Report
The authors have addressed my comments.
Author Response
Thank you.